# Biosynthesis of Pteridines in Insects: A Review

**DOI:** 10.3390/insects15050370

**Published:** 2024-05-19

**Authors:** Juan Ferré

**Affiliations:** Institute of Biotechnology and Biomedicine (BIOTECMED), Universitat de València, 46100 Burjassot, Spain; juan.ferre@uv.es; Tel.: +34-963-544-506

**Keywords:** pterin, lumazine, insect pigments, xanthine dehydrogenase, dihydropterin deaminase, sepiapterin reductase, dihydropterin oxidase, drosopterins, erythropterin, xanthopterin

## Abstract

**Simple Summary:**

Pigments in insects have attracted the interest of scientists since the end of the 19th century. Most yellow, orange or red pigments belong to a family of pteridines which, in addition to their color, are fluorescent under UV light. The fluorescence of colorless intermediates in the pathway can help interspecific recognition. This review discusses, and integrates into one metabolic pathway, the different branches which lead to the synthesis of the pigments commonly found in the wings of butterflies and eyes and bodies of insects. Despite their function as pigments, some pteridine derivatives are important enzyme cofactors in all living organisms.

**Abstract:**

Pteridines are important cofactors for many biological functions of all living organisms, and they were first discovered as pigments of insects, mainly in butterfly wings and the eye and body colors of insects. Most of the information on their structures and biosynthesis has been obtained from studies with the model insects *Drosophila melanogaster* and the silkworm *Bombyx mori*. This review discusses, and integrates into one metabolic pathway, the different branches which lead to the synthesis of the red pigments “drosopterins”, the yellow pigments sepiapterin and sepialumazine, the orange pigment erythropterin and its related yellow metabolites (xanthopterin and 7-methyl-xanthopterin), the colorless compounds with violet fluorescence (isoxanthopterin and isoxantholumazine), and the branch leading to tetrahydrobiopterin, the essential cofactor for the synthesis of aromatic amino acids and biogenic amines.

## 1. Introduction

Pteridines are a family of compounds, some of them acting as pigments, which are widely distributed in the animal kingdom [1]. They were first discovered and studied in butterflies, as pigments in the wing scales, as early as the end of the 19th century (reviewed by Andrade and Carneiro) [1]. They possess functions including visual screening pigments, external signaling, nitrogen excretory substances, essential cofactors of many metabolic reactions, the hydroxylation of aromatic amino acids, and producers of nitric oxide, among others [2]. The accumulation of some pteridines has also been used for age determination in some insect species [3,4,5].

Pteridines take their name from the heterocyclic pteridine ring, which is composed of fused pyrimidine and pyrazine rings (Figure 1). All natural pteridines have a hydroxyl group at position 4. An amino group or a hydroxyl group at position 2 determines whether they belong to the pterin or to the lumazine family, respectively. The hydroxyl groups readily tautomerize into the most stable keto tautomeric form.

Color production in insects can be either pigmentary or structural. Some pteridines contribute to coloration as true pigments because they absorb light in the visual spectrum, whereas others contribute structurally by reflecting or diffracting incident light in ordered structures. Most pteridines emit fluorescence under UV light, a property that assists their purification and identification. The fluorescence of colorless pteridines can also contribute to species/sex identification; Table 1 shows the actual color and the fluorescence color under UV light of some of them.

Most of the information available on the structure and biosynthesis of pteridines has been obtained from the two model insects *Drosophila melanogaster* and the silkworm *Bombyx mori* [1,6] (Figure 2). The fruit fly *D. melanogaster*, with its many eye-color mutants, has contributed enormously to unraveling the pteridine pathway, mainly the branch leading to red eye pigments [7,8]. However, despite all the efforts, the information is still partial or fragmented in some branches, and some metabolic steps are still unknown. In this review, a comprehensive pathway integrating all the available information is proposed for the most common pteridines, the enzymes and their genetic loci when known. This review has been structured into sections for discussion of the metabolic branches leading to the different end products.

## 2. The Pathway Backbone

It has long been established that the precursor of pteridines in all organisms is the nucleotide guanosine-5′-triphosphate (GTP). A series of steps give rise to the intermediates from which the rest of branches of the pathway originate, giving rise to the colored pteridines and important biochemical cofactors. This is what I have called “the pathway backbone” in this review (Figure 3).

GTP is converted to 7,8-dihydroneopterin triphosphate by the action of the enzyme GTP cyclohydrolase I [9,10]. In *D*. *melanogaster*, the structural gene for this enzyme was identified as *Punch* (*Pu*), located at the 57C7-57C8 region of the cytological map of the second chromosome [11]. Two five-member rings open and form a six-member pyrazine ring with the release of formic acid, giving rise to the first pterin in the pathway, 7,8-dihydroneopterin triphosphate. This compound is converted to 6-pyruvoyl-5,6,7,8-tetrahydropterin by the action of the enzyme pyruvoyl-tetrahydropterin synthase, formerly known as sepiapterin synthase A [12,13,14,15]. The *purple* (*pr*) gene, located at 38B3 on the second chromosome in *D. melanogaster*, is the structural gene of this enzyme [16].

The steps leading from 6-pyruvoyl-tetrahydropterin to dihydropterin have not been completely unraveled. A “side-chain-releasing enzyme” was found in *D. melanogaster* that released the side chain, giving rise to 7,8-dihydropterin [17], presumably through the 5,6,7,8-tetrahydropterin intermediate. This tetrahydro intermediate was found, for the first time, in *D. melanogaster* by HPLC with electrochemical detection [18]. The *D. melanogaster* eye-color mutant *Henna^r3^* (*Hn^r3^*) lacked this compound and its derivatives, such as the red-orange “drosopterins”, though not the other pteridines derived from 6-pyruvoyl-tetrahydropterin, suggesting that it was the gene controlling this step. Later on, it was shown that *Henna*, located at 66A12 on the third chromosome, was the structural gene for phenylalanine hydroxylase in *D. melanogaster* [19,20,21,22]. To date, the role of phenylalanine hydroxylase in this metabolic step is not completely clear.

Finally, the conversion of this tetrahydopterin to its dihydro homolog, 7,8-dihydropterin is thought to be non-enzymatic, due to the ease of oxidation of the tetrahydro forms.

Neopterin, the dephosphorylated and fully oxidized form of 7,8-dihydroneopterin triphosphate, relatively common in many organisms, is thought to be a degradation product of the latter.

## 3. The Drosopterin Branch

The red and red-orange pigments of *D. melanogaster* eyes, also present in animals other than arthropods [1], are a family of structurally related compounds with five aromatic rings and a long delocalized system of double bonds which make them absorb the visible range of light. As with many other authors writing on this family of compounds, I will refer generically to all of them as “drosopterins” (in inverted commas), which will refer globally to drosopterin, isodrosopterin, aurodrosopterin, isoaurodrosopterin, neodrosopterin and “fraction e” [23,24]. Since drosopterin and isodrosopterin are enantiomers differing in the orientation of the two hydrogens in the five-member ring, I will generically refer to them as drosopterins. Similarly, the orientation of the two hydrogens in the five-member ring differentiates the two enantiomers aurodrosopterin and isoaurodrosopterin, and the term aurodrosopterins will be used here to refer to the two of them. Although the chemical structure of the red pigment neodrosopterin is known, there is no knowledge of the metabolic steps required to synthesize it. Neither the structure nor the biosynthesis of the so-called “fraction e” are known.

It has been shown that, in vitro, drosopterins and aurodrosopterins can be synthesized by the spontaneous condensation of a pyrimidodiazepine intermediate with 7,8-dihydropterin and 7,8-dihydrolumazine, respectively [24,25] (Figure 4). It is not known whether these condensation reactions also take place spontaneously in vivo or if they are enzymatically catalyzed. 6-Pyruvoyl-tetrahydropterin is converted to the pyrimidodiazepine by the enzyme pyrimidodiazepine synthase, which is encoded by the *sepia* (*se*) gene located at 66D5 on the third chromosome in *D. melanogaster* [26,27]. The pyrimidodiazepine intermediate was also described as the “quench spot” because this compound, once separated by cellulose thin-layer chromatography, quenched the UV light; however, at very low temperatures (e.g., when liquid nitrogen was poured on the cellulose plate), the compound showed bright green fluorescence [28,29,30]. It was also referred to as acetyldihydrohomopterin and it was very conspicuous in the *D. melanogaster* eye-color mutant *Hn^r3^* because of its accumulation due to the blocking of the steps leading to 7,8-dihydropterin [7].

7,8-Dihydrolumazine, required for the synthesis of aurodrosopterins, is derived from its pterin counterpart 7,8-dihydropterin by the enzyme dihydropterin deaminase [31,32]. Although this enzyme also accepts guanine as a substrate, it is highly specific for 7,8-dihydropterin since it does not accept other pteridines as substrates, including sepiapterin and isoxanthopterin [32]. The *D. melanogaster* gene coding for this enzyme has been identified as the CG18143 gene, which is located at 82A1 on the third chromosome [31,32].

## 4. The Sepiapterin Branch

6-Pyruvoyl-tetrahydropterin also serves as the precursor of the branch leading to the yellow pigments sepiapterin and sepialumazine (Figure 5). In *D. melanogaster*, the accumulation of sepiapterin is very evident in those eye-color mutants with the drosopterin branch blocked, such as *sepia* and *Henna^r3^* [7,28]. Both mutants have very dark eyes due to a lack or very reduced levels of “drosopterins” and the accumulation of sepiapterin. The excess of 6-pyruvoyl-tetrahydropterin is then diverted to the sepiapterin synthesis and also to the less abundant yellow compound 2′-deoxysepiapterin (also known as isosepiapterin). 2′-Deoxysepiapterin has been found in the Mexican fruit fly and has been proposed as an age determination marker [3,4].

The synthesis of sepiapterin from 6-pyruvoyl-tetrahydropterin in *Drosophila* is catalyzed by the enzyme pyruvoyl-tetrahydropterine reductase, formerly known as sepiapterin synthase B [13,15,33]. This enzyme converts 6-pyruvoyl-tetrahydropterin to metastable 6-lactoyltetrahydropterin, which is autoxidized to sepiapterin under aerobic conditions.

In some insect species, though not in *D. melanogaster*, sepiapterin can be converted to its yellow counterpart sepialumazine by the enzyme sepiapterin deaminase [34,35]. Sepialumazine is the most common yellow pigment found in the larvae of the silkworm *Bombyx mori* [36,37,38]. The *lemon* (*lem*) mutant of this species, with larvae of a yellow color instead of the typical grey color, has impaired sepiapterin reductase activity and, as a consequence, accumulates high amounts of sepialumazine and sepiapterin in the larva integument [39].

## 5. The Isoxanthopterin Branch

The oxidation of 7,8-dihydropterin to pterin was shown to be catalyzed by the enzyme dihydropterin oxidase. This enzyme was first purified from *D. melanogaster* and, though it can use many 7,8-dihydropterin compounds as substrates, the biochemical parameters indicated that its physiological role in *D. melanogaster* was to participate in the synthesis of isoxanthopterin [40,41] (Figure 6). Among all the *D. melanogaster* eye-color mutants analyzed, only *little isoxanthopterin* (*lix*) did not show any detectable dihydropterin oxidase activity [42]. Biochemical and genetic analyses showed that the *lix* locus, at the 7D10-7F1-2 segment of the X chromosome, contained the structural gene of dihydropterin oxidase [42].

The conversion of pterin into isoxanthopterin involves the hydroxylation of position 7 of the pteridine ring, and this is accomplished by the enzyme xanthine dehydrogenase [43,44,45]. The structural gene for this enzyme in *D. melanogaster* is *rosy* (*ry*), located at the 87DE region on the third chromosome [46]. Isoxanthopterin is not found as a screening pigment in the eyes since it is not found in *D. melanogaster* heads, but in other parts of the body and mainly in the testis sheath [7,47]. Because of its body localization, its ubiquity in insect species, and the fact that its accumulation is not synchronized with eye pteridines, it has been proposed that isoxanthopterin may be a “storage–excretion” form of nitrogen in insects.

Isoxantholumazine, also known as violapterin, is found in Lepidoptera and Hemiptera, but not in *Drosophila* [48,49,50]. Its synthesis from isoxanthopterin is catalyzed by the enzyme isoxanthopterin deaminase [50,51]. This enzyme is very specific for isoxanthopterin and does not accept other pterins as substrates, including 7,8-dihydropterin [50].

## 6. The Xanthopterin Branch

This branch leads to the yellow and orange pigments containing the keto group at position 6 on the pteridine ring (Figure 7). An early study in the butterfly *Colias eurytheme*, tracking the incorporation of radioactively labeled precursors into the pteridine ring, showed that xanthopterin is the precursor of leucopterin and erythropterin [52].

The hydroxylation of 7,8-dihydropterin at position 6 is accomplished by the enzyme xanthine dehydrogenase [45]. This reaction does not take place in *D. melanogaster* despite the fact that xanthine dehydrogenase is found in this species and it is responsible for the synthesis of isoxanthopterin [7]. The explanation for this apparent paradox may be the occurrence of xanthine dehydrogenase isozymes [53], differences in the compartmentation of substrates or enzymes in granules, cells or tissues, and/or differences in the kinetics of the reactions (either enzymatic or spontaneous) using 7,8-dihydropterin as the substrate.

The oxidation of 7,8-dihydroxanthopterin to the yellow pigment xanthopterin is most likely catalyzed by the enzyme dihydropterin oxidase. This enzyme was first purified from *D. melanogaster* and was shown to be able to use many 7,8-dihydropterin compounds as substrates [40,41].

Xanthopterin is also a substrate of xanthine dehydrogenase, and is converted to the colorless compound leucopterin [45], which functions as a white pigment in the wings of butterflies [54].

The conversion of xanthopterin into the orange pigment erythropterin involves the addition of a lateral chain at position 7, possibly from pyruvate, malate or lactate [52]. It has been proposed that this step is catalyzed by the enzyme erythropterin synthase [6], though at the time of this review no studies on this enzyme have been found. The yellow pigment 7-methylxanthopterin, also known as chrysopterin, is thought to be a degradation product of erythropterin. Erythropterin, along with the yellow pigments xanthopterin and 7-methylxanthopterin, are the main pteridine pigments in Hemiptera [48,49,55,56,57,58,59] (a compilation of pteridines found in Hemiptera can be found in Table S4 of the Suppl. Meth. of Vargas-Lowman et al.) [59].

## 7. The Tetrahydrobiopterin Branch

5,6,7,8-Tetrahydrobiopterin is a key cofactor in the metabolism of animals, some bacteria and fungi, and insects in particular [2,60]. This important cofactor has a complex metabolism with two biosynthetic pathways: the *de novo* pathway and the *salvage* pathway [61] (Figure 8). There is still a third pathway, not directly related to its synthesis but to its recovery, which allows 5,6,7,8-tetrahydrobipterin to regenerate after its participation as a cofactor in enzyme reactions: the *recycling* pathway [60].

Sepiapterin reductase is the key enzyme in the *de novo* and *salvage* pathways and it was initially discovered in *B. mori* [62,63]. Given the importance of 5,6,7,8-tetrahydrobiopterin in the metabolism, it is not surprising that, among the many *D. melanogaster* eye-color mutants analyzed, one has never been found that lacks sepiapterin reductase activity [64]. In *B. mori*, *lemon lethal* (*lem^l^*) is a homozygous lethal allele due to the lack of sepiapterin reductase [39].

The *de novo* pathway involves the enzymes pyruvoyl-tetrahydropterin reductase and sepiapterin reductase [33]. However, the participation of another enzyme cannot be discounted, since in a study carried out in *D. melanogaster* in which sepiapterin reductase was almost completely depleted, the levels of 5,6,7,8-tetrahydropterin were only decreased to 50%, suggesting the participation of other enzymes in 5,6,7,8-tetrahydrobiopterin biosynthesis [65]. In mammals, it was shown that, in addition to sepiapterin reductase, the synthesis of 5,6,7,8-tetrahydrobiopterin can be accomplished by an enzyme of the aldose reductase family (to which pyruvoyl-tetrahydropterin reductase belongs) and a carbonyl reductase [66,67]. Therefore, it is possible that in insects, the steps catalyzed by sepiapterin reductase could also be performed by other enzymes.

The *salvage* pathway leads to 5,6,7,8-tetrahydrobiopterin from the precursor sepiapterin, formed by the spontaneous oxidation of 6-lactoyltetrahydropterin. Sepiapterin reductase (and/or a carbonyl reductase) converts it into 7,8-dihydrobiopterin which, in turn, is reduced to 5,6,7,8-tetrahydrobiopterin by the enzyme dihydrofolate reductase [61]. 7,8-Dihydrobiopterin can otherwise be converted to biopterin by the enzyme dihydropterin oxidase (encoded by the *lix* locus in *D. melanogaster*) [42].

## 8. Conclusions

Pteridines are important cofactors for many biological functions of all living organisms and contribute to insect pigmentation, along with other pigments. GTP is the precursor of all pteridines. However, the biosynthetic pathway is not linear, with many branches leading to different end products which are active or not depending on the insect order or species within it. A proposed comprehensive pathway for the biosynthesis of pteridines, which integrates the pathways occurring in different insect species, is shown in Figure 9. The enzymes controlling each step and their genetic loci in *D. melanogaster* are also shown. Following the style in preceding sections, the pathway has been compartmentalized according to the different end products.

The first steps from GTP to 7,8-dihydropterin, named “the pathway backbone” in this review, give rise to the two key intermediates from which the rest of branches of the pathway originate: 6-pyruvoyl-tetrahydropterin and 7,8-dihydropterin.

The branch leading to the orange-red eye pigments “drosopterins” (drosopterin, isodrosopterin, aurodrosopterin and isoaurodrosopterin) takes 6-pyruvoyl-tetrahydropterin and 7,8-dihydropterin as starting points from “the pathway backbone”. Since the conversion of the former to pyrimidodiazepine is an obligate step in the synthesis of “drosopterins”, this branch will be absent in insect species lacking the pirimidodiazepine synthase enzyme. This enzyme would act as a “switch” for the drosopterin branch.

The branch leading to the yellow pigments sepiapterin and sepialumazine starts from 6-pyruvoyl-tetrahydropterin. Sepiapterin is rather ubiquitous since it is formed by the spontaneous oxidation of an intermediate in the *salvage* pathway of 5,6,7,8-tetrahydrobiopterin. However, the presence or absence of the sepiapterin deaminase enzyme is key for the synthesis of sepialumazine, which is found in *Bombyx mori* but not in *Drosophila*.

The colorless compounds with violet fluorescence (isoxanthopterin and isoxantholumazine) originate from 7,8-dihydropterin. The role of these two compounds is not clear and it is possible that they could act as “storage–excretion” forms of nitrogen in insects. The enzyme isoxanthopterin deaminase acts as a “switch” for the synthesis of isoxantholumazine, also known as violapterin. This compound is found in Lepidoptera and Hemiptera, but not in *Drosophila*.

The “xanthopterin branch” leads to the yellow and orange pigments found in many Lepidoptera and Hemiptera. Leucopterin, a colorless derivative of xanthopterin, has been shown to be responsible for the white color of butterfly wings. This branch starts from 7,8-dihydropterin through the action of xanthine dehydrogenase, which adds a hydroxyl group at position 6 of the pteridine ring. Since this enzyme also catalyzes the synthesis of isoxanthopterin and leucopterin, in which the hydroxyl group is added at position 7, it is thought that different isozymes could be involved.

6-Pyruvoyl-tetrahydropterin is the starting point from “the pathway backbone” for the synthesis of 5,6,7,8-tetrahydrobiopterin. Its synthesis can proceed through two pathways: the *de novo* pathway and the *salvage* pathway, the latter through sepiapterin and 7,8-dihydrobiopterin. The “tetrahydrobiopterin branch” is found in all insect species, since 5,6,7,8-tetrahydrobiopterin is an essential cofactor for many reactions, the synthesis of aromatic amino acids and biogenic amines among them.

There are still many aspects of pteridine metabolism to be understood. Given that most information has been generated from studies on *D. melanogaster* and *B. mori*, it would be desirable to expand these studies to other non-model species. Advances in genomics, proteomics, transcriptomics, metabolomics and liquid/mass spectrometry can facilitate this task. Improved knowledge of pteridine metabolism will undoubtedly help us to better understand their role in external signaling and their interactions with the general metabolism and physiology in insects.

## Figures and Tables

**Figure 1 insects-15-00370-f001:**
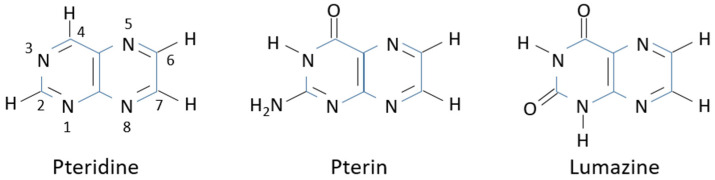
Structures of simple pteridines which give rise to all pteridine derivatives.

**Figure 2 insects-15-00370-f002:**
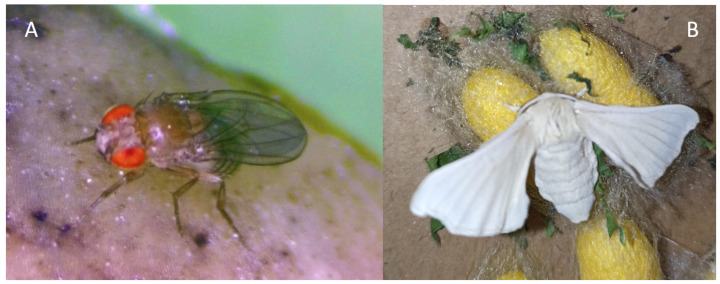
The two insects that most contributed to the biochemistry of pteridines. (**A**) The fruit fly *Drosophila melanogaster* (the red eye color is due to drosopterins). (**B**) Moth of the silkworm *Bombyx mori* and some cocoons.

**Figure 3 insects-15-00370-f003:**
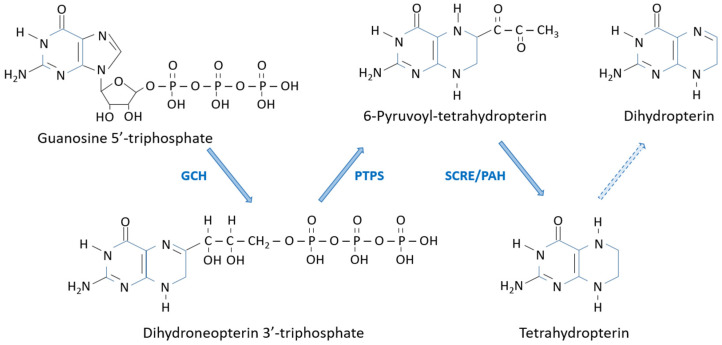
Biosynthesis of 6-pyruvoyl-tetrahydropterin and 7,8-dihydropterin from which the rest of the pathway branches originate. GCH: GTP cyclohydrolase I; PAH: Phenylanaline hydroxylase; PTPS: Pyruvoyl-tetrahydropterin synthase; SCRE: Side-chain-releasing enzyme. Dashed line: spontaneous?

**Figure 4 insects-15-00370-f004:**
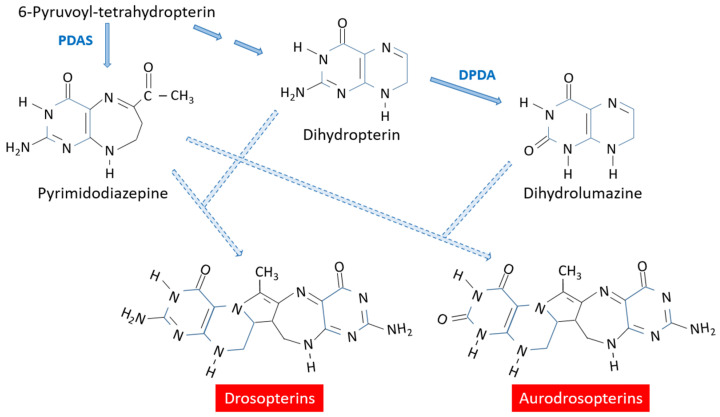
Biosynthesis of the orange-red pigments drosopterins and aurodrosopterins. DPDA: Dihydropterin deaminase; PDAS: Pyrimidodiazepine synthase. Dashed lines: spontaneous?

**Figure 5 insects-15-00370-f005:**
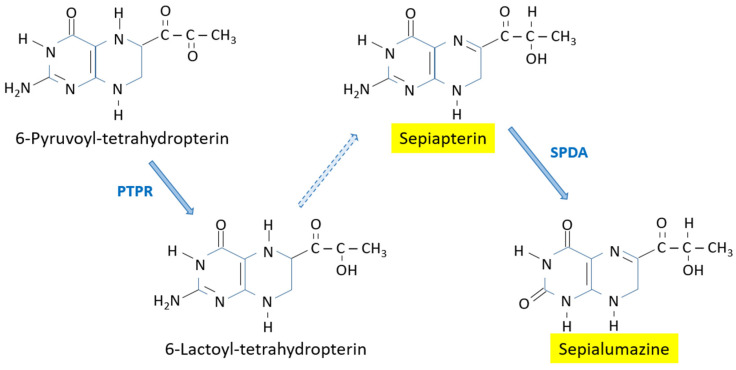
Biosynthesis of the yellow pigments sepiapterin and sepialumazine. PTPR: Pyruvoyl-tetrahydropterin reductase; SPDA: Sepiapterin deaminase. Dashed line: spontaneous?

**Figure 6 insects-15-00370-f006:**
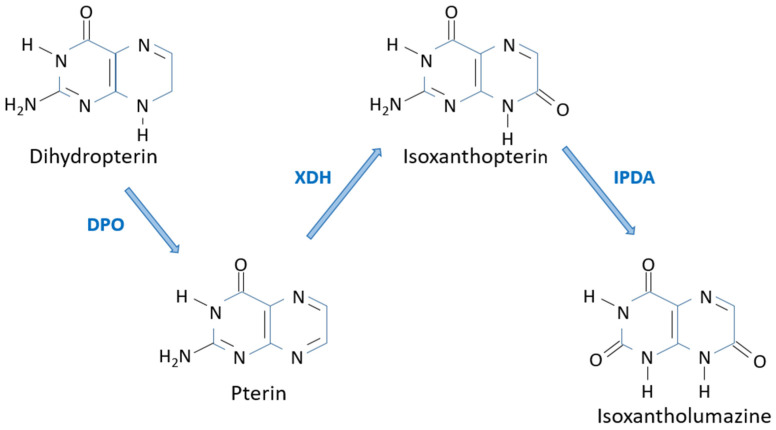
Biosynthesis of the colorless violet-fluorescent isoxanthopterin and isoxantholumazine. DPO: Dyhydropterin oxidase; IPDA: Isoxanthopterin deaminase; XDH: Xanthin dehydrogenase.

**Figure 7 insects-15-00370-f007:**
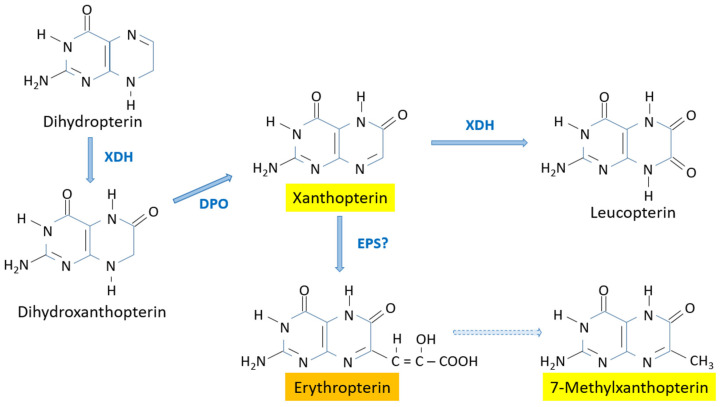
Biosynthesis of the yellow and orange pigments of the xanthopterin branch. DPO: Dyhydropterin oxidase; EPS?: Erythropterin synthase (proposed); XDH: Xanthin dehydrogenase. Dashed line: spontaneous?

**Figure 8 insects-15-00370-f008:**
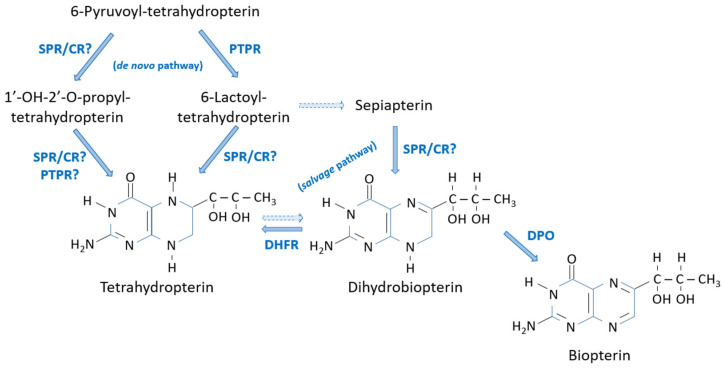
Biosynthesis of tetrahydrobiopterin by *de novo* and *salvage* pathways. CR?: carbonyl reductase (proposed); DHFR: Dihydrofolate reductase; DPO: Dyhydropterin oxidase; PTPR: Pyruvoyl-tetrahydropterin reductase; SPR: Sepiapterin reductase. Dashed lines: spontaneous?

**Figure 9 insects-15-00370-f009:**
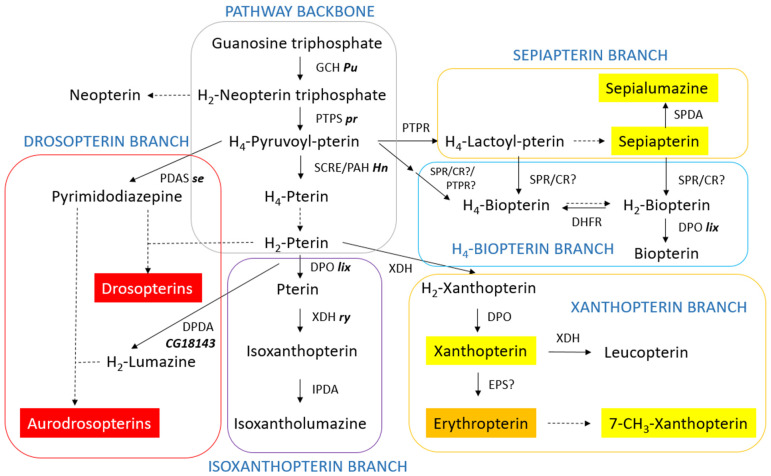
Proposed comprehensive pteridine pathway in insects. CR?: carbonyl reductase (proposed); DHFR: Dihydrofolate reductase; DPDA: Dihydropterin deaminase; DPO: Dyhydropterin oxidase; EPS?: Erythropterin synthase (proposed); GCH: GTP cyclohydrolase I; IPDA: Isoxanthopterin deaminase; PAH: Phenylanaline hydroxylase; PDAS: Pyrimidodiazepine synthase; PTPR: Pyruvoyl-tetrahydropterin reductase; PTPS: Pyruvoyl-tetrahydropterin synthase; SCRE: Side-chain-releasing enzyme; SPDA: Sepiapterin deaminase; SPR: Sepiapterin reductase; XDH: Xanthin dehydrogenase. Genes coding for their respective enzymes in *Drosophila melanogaster* are written in bold phase to the right of the enzyme abbreviation. Dashed lines: spontaneous?

**Table 1 insects-15-00370-t001:** Physical characteristics of the most common pteridines found in insects.

Name	Color	Fluorescence
Drosopterins	orange-red	orange-red
Aurodrosopterins	orange-red	orange-red
Erythropterin	orange	orange
Xanthopterin	yellow	yellow-green
7-Methyl-xanthopterin	yellow	yellow-green
Sepiapterin	yellow	yellow
Sepialumazine	yellow	yellow
Neopterin	colorless	blue
Biopterin	colorless	blue
Pterin	colorless	blue
Leucopterin	colorless	pale blue
Isoxanthopterin	colorless	violet
Isoxantholumazine	colorless	violet

## Data Availability

This article has no additional data.

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
