# Peer review of "Biosynthesis of Pteridines in Insects: A Review"

_insects, 2024, doi:10.3390/insects15050370_

Round 1

Reviewer 1 Report

Comments and Suggestions for Authors

After carefully reviewing the manuscript titled "Biosynthesis of Pteridines in Insects: A Review," and found it to be of great interest. The review delves into the intricate world of pteridine biosynthesis, shedding light on their multifaceted roles across various organisms. Beginning with their discovery in butterflies and spanning through their functions as pigments, signaling molecules, and metabolic cofactors, the review provides a comprehensive overview of these fascinating compounds.

Overall, the review presents a well-structured narrative, offering valuable insights into the diverse aspects of pteridine biology. However, I have a few inquiries and suggestions to enhance clarity and provide further depth to certain aspects of the review.

Introduction & Pathway backbone

1. In the first paragraph, there is a small typo with "as earlier as" which should be corrected to "as early as."

2. When discussing the pathway backbone, there's a mention of "formarly known as sepiapterin synthase A." I believe this should be corrected to "formerly known as sepiapterin synthase A."

5. In the sentence "Neopterin, the dephosphorylated and fully oxidized form of 7,8-dihydroneopterin triphosphate, which is found relatively common in many organisms," there's a grammatical issue with "relatively common." It should be corrected to "relatively common in many organisms."

6. Throughout the text, make sure to maintain consistent formatting and spelling conventions for scientific terms and symbols, such as "GTP" and "pteridine."

The drosopterin branch

Towards the end of the paragraph, "Neither the structure nor the biosynthesis is known on the so called 'fraction e." It seems like there's a missing word or a typo. Could you clarify the intended meaning or correct the typo?

In the second paragraph, "It is not known whether these condensation reactions also take place spontaneously in vivo or if they are enzymatically catalized." There's a typo with "catalized" which should be corrected to "catalyzed."

In the same sentence, there's a usage issue with "enzymatically catalized." It might be clearer to say "enzymatically catalyzed in vivo."

In the same sentence, "The D. melanogaster gene coding for this enzyme has been identified as CG18143, located at 82A1 on the third chromosome," it might be clearer to specify that "CG18143" refers to the gene symbol or identifier.

Consider providing a brief explanation or context for terms like "sepia" and "Hennar" to aid readers who might not be familiar with these terms in the context of Drosophila genetics.

In the same sentence, “which is codified by the sepia (se) gene located at 66D5 on the third chromosome in D. melanogaster." It should be corrected to "also encoded".

The sepiapterin branch

Towards the end of the paragraph, "accumulate high amounts of sepialumazine and sepiapterin in the larva integument," it seems like there's a plural agreement issue with "accumulate" and "amounts." It should be corrected to "accumulates high amounts of sepialumazine and sepiapterin in the larval integument."

Consider providing a brief explanation for terms like "lemon (lem) mutant" to aid readers who might not be familiar with these terms in the context of silkworm genetics.

The isoxanthopterin branch

In the sentence "The oxidation of 7,8-dihydropterin to pterin was shown to be catalized by the enzyme dihydropterin oxidase," there's a typo with "catalized," which should be corrected to "catalyzed."

Additionally, consider rephrasing "7,8-dihydropterin among them" to "including 7,8-dihydropterin," for clarity and smoother flow of the sentence.

The xanthopterin branch

"The conversion of pterin into isoxanthopterin involves hydroxylation of the 7 position of the pteridine ring and this is accomplished by the enzyme xanthine dehydrogenase." - "the 7 position" should be corrected to "position 7."

"This reaction does not take place in D. melanogaster despite the fact that xanthine dehydrogenase is found in this species as responsible for the synthesis of isoxanthopterin." - "as responsible for" should be corrected to "responsible for."

"This enzyme was first purified from D. melanogaster and shown to be able to use many 7,8-dihydropterins as substrates." - "dihydropterins" should be corrected to "dihydropterin" for consistency.

The explanation for this apparent paradox may be the occurrence of different xanthine dehydrogenase isozymes" - "different xanthine dehydrogenase isozymes" could be revised for clarity to "variations in xanthine dehydrogenase isozymes."

"This branch leads to the yellow and orange pigments containing the keto group at position of the pteridine ring." - "of the pteridine ring" should be corrected to "on the pteridine ring" for clarity.

That xanthopterin is the precursor of leucopterin and erythropterin was shown by incorporation of radioactively labelled precursors in the butterfly Colias eurytheme." - "That" could be replaced with "The fact that" for clearer expression.

"The hydroxylation of 7,8-dihydropterin at the 6 position is accomplished by the enzyme xanthine dehydrogenase." - "the 6 position" should be corrected to "position 6."

The tetrahydrobiopterin branch

"Sepiapterin reductase is the key enzyme in the de novo and salvage pathways and it was first discovered in B. mori." - "it was first discovered" could be revised for clarity to "was initially discovered."

"Given the importance of 5,6,7,8-tetrahydrobiopterin in the metabolism, it is not strange that, among the many D. melanogaster eye-color mutants analyzed, it has never been found one lacking sepiapterin reductase activity." - This sentence could be rephrased for clarity and grammatical accuracy.

"However, the participation of another enzyme cannot be discarded since in a study carried out in D. melanogaster…." - "discarded" should be corrected to "discounted."

"Therefore, it is possible that also in insects the steps catalysed by sepiapterin reductase could also be performed by other enzymes." - "catalysed" should be corrected to "catalyzed."

"The salvage pathway leads to 5,6,7,8-tetrahydrobiopterin from the precursor sepiapterin, formed by the spontaneous oxidation of 6-lactoyltetrahydropterin." - "spontaneous" could be replaced with "spontaneously" for clarity.

"Sepiapterin reductase (and/or a carbonyl reductase) converts it into 7,8-dihydrobiopterin which, in turn, can be reduced to 5,6,7,8-tetrahydrobiopterin by the enzyme dihydrofolate reductase." - "can be reduced to" could be revised for clarity to "is reduced to."

These corrections should help improve the clarity and accuracy of the text. Consider breaking down complex sentences into smaller ones for easier comprehension, especially when explaining intricate biochemical pathways. This can improve the readability and clarity of the text.

Comments on the Quality of English Language

none

Author Response

Reviewer 1:

Overall, the review presents a well-structured narrative, offering valuable insights into the diverse aspects of pteridine biology. However, I have a few inquiries and suggestions to enhance clarity and provide further depth to certain aspects of the review.

Introduction & Pathway backbone

  1. In the first paragraph, there is a small typo with "as earlier as" which should be corrected to "as early as."

Done 

  1. When discussing the pathway backbone, there's a mention of "formarly known as sepiapterin synthase A." I believe this should be corrected to "formerly known as sepiapterin synthase A."

 Done

  1. In the sentence "Neopterin, the dephosphorylated and fully oxidized form of 7,8-dihydroneopterin triphosphate, which is found relatively common in many organisms," there's a grammatical issue with "relatively common." It should be corrected to "relatively common in many organisms."

 Done

  1. Throughout the text, make sure to maintain consistent formatting and spelling conventions for scientific terms and symbols, such as "GTP" and "pteridine."

 The term “pteridine” is used generically, when referring to both pterins and lumazines. I’ve not found any inconsistence with GTP.

The drosopterin branch

Towards the end of the paragraph, "Neither the structure nor the biosynthesis is known on the so called 'fraction e." It seems like there's a missing word or a typo. Could you clarify the intended meaning or correct the typo?

 Done

In the second paragraph, "It is not known whether these condensation reactions also take place spontaneously in vivo or if they are enzymatically catalized." There's a typo with "catalized" which should be corrected to "catalyzed."

It has been corrected throughout the manuscript. 

In the same sentence, there's a usage issue with "enzymatically catalized." It might be clearer to say "enzymatically catalyzed in vivo."

 It is already explicitated in the sentence “…also take place spontaneously in vivo or if they are enzymatically catalyzed.” Adding in vivo after catalysed would be redundant.

In the same sentence, "The D. melanogaster gene coding for this enzyme has been identified as CG18143, located at 82A1 on the third chromosome," it might be clearer to specify that "CG18143" refers to the gene symbol or identifier.

As a matter of fact, the authors of the original paper refer to it as THE GENE. Textually, they write: “To identify the structural gene, the purified enzyme was analyzed by MALDI-TOF mass spectrometry. Thirty-seven peptides (range, 1000–3000 Da) were analyzed by a data base search that identified the enzyme as the product of the Drosophila CG18143 gene, located at the 82A1 region of the cytological map of chromosome 3R (GenBankTM accession number AE003607; NCBI RefSeq accession number NP_649439)”.

Consider providing a brief explanation or context for terms like "sepia" and "Hennar" to aid readers who might not be familiar with these terms in the context of Drosophila genetics.

A sentence describing the phenotype has been added at the next section where the name of the two mutants appear. 

In the same sentence, “which is codified by the sepia (se) gene located at 66D5 on the third chromosome in D. melanogaster." It should be corrected to "also encoded".

 Done.

The sepiapterin branch

Towards the end of the paragraph, "accumulate high amounts of sepialumazine and sepiapterin in the larva integument," it seems like there's a plural agreement issue with "accumulate" and "amounts." It should be corrected to "accumulates high amounts of sepialumazine and sepiapterin in the larval integument."

Done. 

Consider providing a brief explanation for terms like "lemon (lem) mutant" to aid readers who might not be familiar with these terms in the context of silkworm genetics.

To clarify in which the lem mutation affect the phenotype, I have now included that the wild type phenotype is grey, to indicate the effect of the mutation. The sentence is now “The lemon (lem) mutant of this species, with larvae of a yellow color instead of the typical grey color, has impaired sepiapterin reductase activity and, as a consequence, accumulates high amounts of sepialumazine and sepiapterin in the larva integument [39].”

The isoxanthopterin branch

In the sentence "The oxidation of 7,8-dihydropterin to pterin was shown to be catalized by the enzyme dihydropterin oxidase," there's a typo with "catalized," which should be corrected to "catalyzed."

 Done.

Additionally, consider rephrasing "7,8-dihydropterin among them" to "including 7,8-dihydropterin," for clarity and smoother flow of the sentence.

 Done.

The xanthopterin branch

"The conversion of pterin into isoxanthopterin involves hydroxylation of the 7 position of the pteridine ring and this is accomplished by the enzyme xanthine dehydrogenase." - "the 7 position" should be corrected to "position 7."

This has been corrected in this an other parts of the manuscript where the number preceded the word “position”. 

"This reaction does not take place in D. melanogaster despite the fact that xanthine dehydrogenase is found in this species as responsible for the synthesis of isoxanthopterin." - "as responsible for" should be corrected to "responsible for."

 This part of the sentence has been changed.

"This enzyme was first purified from D. melanogaster and shown to be able to use many 7,8-dihydropterins as substrates." - "dihydropterins" should be corrected to "dihydropterin" for consistency.

The plural in dihydropterins referred to many compounds with this structure. Now the sentence has been clarified by saying “7,8-dihydropterin compounds”. 

The explanation for this apparent paradox may be the occurrence of different xanthine dehydrogenase isozymes" - "different xanthine dehydrogenase isozymes" could be revised for clarity to "variations in xanthine dehydrogenase isozymes."

I do not consider this change appropriate because the sentence means that different isozymes could be responsible for the different reactions in different species. 

"This branch leads to the yellow and orange pigments containing the keto group at position of the pteridine ring." - "of the pteridine ring" should be corrected to "on the pteridine ring" for clarity.

Done. 

That xanthopterin is the precursor of leucopterin and erythropterin was shown by incorporation of radioactively labelled precursors in the butterfly Colias eurytheme." - "That" could be replaced with "The fact that" for clearer expression.

The sentence has been rephrased. 

"The hydroxylation of 7,8-dihydropterin at the 6 position is accomplished by the enzyme xanthine dehydrogenase." - "the 6 position" should be corrected to "position 6."

Done. 

The tetrahydrobiopterin branch

"Sepiapterin reductase is the key enzyme in the de novo and salvage pathways and it was first discovered in B. mori." - "it was first discovered" could be revised for clarity to "was initially discovered."

Done 

"Given the importance of 5,6,7,8-tetrahydrobiopterin in the metabolism, it is not strange that, among the many D. melanogaster eye-color mutants analyzed, it has never been found one lacking sepiapterin reductase activity." - This sentence could be rephrased for clarity and grammatical accuracy.

It has been changed. 

"However, the participation of another enzyme cannot be discarded since in a study carried out in D. melanogaster…." - "discarded" should be corrected to "discounted."

Done. 

"Therefore, it is possible that also in insects the steps catalysed by sepiapterin reductase could also be performed by other enzymes." - "catalysed" should be corrected to "catalyzed."

Done. 

"The salvage pathway leads to 5,6,7,8-tetrahydrobiopterin from the precursor sepiapterin, formed by the spontaneous oxidation of 6-lactoyltetrahydropterin." - "spontaneous" could be replaced with "spontaneously" for clarity.

 I discussed with an English expert and confirmed that “spontaneous oxidation” is more appropriate than “spontaneously oxidation”.

"Sepiapterin reductase (and/or a carbonyl reductase) converts it into 7,8-dihydrobiopterin which, in turn, can be reduced to 5,6,7,8-tetrahydrobiopterin by the enzyme dihydrofolate reductase." - "can be reduced to" could be revised for clarity to "is reduced to."

Done.

These corrections should help improve the clarity and accuracy of the text. Consider breaking down complex sentences into smaller ones for easier comprehension, especially when explaining intricate biochemical pathways. This can improve the readability and clarity of the text.

I’m very grateful with all the reviewer considerations, especially with his/her English corrections. 

Reviewer 2 Report

Comments and Suggestions for Authors

The manuscript by Juan Ferré reviews the biosynthesis of pteridines in insects. Pteridines are biosynthesized compounds that insects use/have as pigments and/or fluorescent molecules, etc. Most of the known information has been obtained from two model insects Drosophila melanogaster and bombyx mori. The review is well-written, relevant to this journal,  very informative, good references. The author has done an excellent job putting this together. It merits publications almost as it is.

Minor revisions, 

1. Some of the chemical structures are missing some atoms, it may be due to the conversion to PDF, but sometimes I see H2, where it should be NH2, (many occasions). I recommend to draw the triphosphate groups instead of P3.

2. It would be nice to have actual pictures of both insects and perhaps an insect's body/diagram of where the pigments are commonly found. 

Author Response

Reviewer 2:

Minor revisions, 

  1. Some of the chemical structures are missing some atoms, it may be due to the conversion to PDF, but sometimes I see H2, where it should be NH2, (many occasions). I recommend to draw the triphosphate groups instead of P3.

I’m very glad with this observation. The way I wanted to indicate two H atoms bound to a C in the ring was misleading, since it seemed that a N was missing before the two H atoms (-NH2), which was not the case. To avoid this misinterpretation, now all figures (except the first one) have been changed using the more standard way to represent the H in the aromatic rings.

The triphosphate groups have been drawn in GTP.

  1. It would be nice to have actual pictures of both insects and perhaps an insect's body/diagram of where the pigments are commonly found. 

A picture of the two insect species has been included, indicating (in the legend) the accumulation of pteridines in the fly eyes.

Reviewer 3 Report

Comments and Suggestions for Authors

The review article, "Biosynthesis of pteridines in insects: A review," outlines the various colored pigments in insects by emphasizing the biosynthesis of Pteridines, focusing on the different research conducted on fruit flies and silk moths. The author explains the biosynthesis pathways in six various sections and concludes the review by proposing a comprehensive pteridine pathway in insects. Overall, the article provides some informative information and tries to expand the pathways' understandings. However, some issues need to be addressed before acceptance.

Major points for improvement: 

The first sentence is a copy of another article, and the reference is missing. Please consider paraphrasing with appropriate reference (Bel et al. 1997).

The introduction mentions various functions of pteridines; it would be helpful to elaborate on the significance of these compounds in organisms' development. Also, adding a brief overview of the recent advancements and gaps in the research area would be beneficial. 

L 30-In my opinion, citing the original discovery article of Pteridines is better than citing a review article. 

The review needs a logical flow between the paragraphs: Table 1 does not fit the article's introduction well. Please consider restructuring sentences or adding proper transitions between paragraphs.

There are some instances where it is unclear about some metabolic steps and enzymatic reactions. 

The use of technical terms and abbreviations sometimes makes it hard to understand the pathways. A glossary of terms/ abbreviations will be helpful.

A concise summary or conclusion section summarizing the major point would be beneficial. It was challenging to understand and follow the proposed pathway. A simple graphical representation could also add value to the review.

Comments on the Quality of English Language

Minor language editing

Author Response

Reviewer 3:

The review article, "Biosynthesis of pteridines in insects: A review," outlines the various colored pigments in insects by emphasizing the biosynthesis of Pteridines, focusing on the different research conducted on fruit flies and silk moths. The author explains the biosynthesis pathways in six various sections and concludes the review by proposing a comprehensive pteridine pathway in insects. Overall, the article provides some informative information and tries to expand the pathways' understandings. However, some issues need to be addressed before acceptance.

Major points for improvement: 

The first sentence is a copy of another article, and the reference is missing. Please consider paraphrasing with appropriate reference (Bel et al. 1997).

The sentence has been rephrased and a reference added.

The introduction mentions various functions of pteridines; it would be helpful to elaborate on the significance of these compounds in organisms' development.

I could not find information on the possible function of pteridines in the development of insects.

Also, adding a brief overview of the recent advancements and gaps in the research area would be beneficial. 

This is already briefly mentioned in the last paragraph of the introduction and then described in detail in the following sections.

L 30-In my opinion, citing the original discovery article of Pteridines is better than citing a review article. 

I also agree in that it is better to cite the original article than a review, but the discovery of pteridines goes back to between 1889 and 1895 and I did not have access to the original publications. This is why I opted for citing the review which cites them.

The review needs a logical flow between the paragraphs: Table 1 does not fit the article's introduction well. Please consider restructuring sentences or adding proper transitions between paragraphs.

The introduction of Table 1 has been changed to make it more logical an appropriated.

There are some instances where it is unclear about some metabolic steps and enzymatic reactions. 

The use of technical terms and abbreviations sometimes makes it hard to understand the pathways. A glossary of terms/ abbreviations will be helpful.

All the abbreviations can be found in the legend of the figures.

A concise summary or conclusion section summarizing the major point would be beneficial. It was challenging to understand and follow the proposed pathway. A simple graphical representation could also add value to the review.

The conclusion section has tried to summarize all major points from each section/branch of the pathway. A graphical abstract has also been provided together with the manuscript.

Round 2

Reviewer 1 Report

Comments and Suggestions for Authors

Thank you for addressing the comments and suggestions I provided on the review titled " Biosynthesis of pteridines in insects: A review".

The authors have provided valuable clarifications.

Author Response

Thanks to the reviewer for the helpful comments

Reviewer 3 Report

Comments and Suggestions for Authors

The revised version is more instructive.

Author Response

(The authors gave the same response as above.)
